# Synthesis of an Aluminum Alloy Metal Matrix Composite Using Powder Metallurgy: Role of Sintering Parameters

Kanhu C. Nayak [1,*], Kedarnath K. Rane [2], Prashant P. Date [3] and T. S. Srivatsan [4]

1    School of Advanced Materials Engineering, Kookmin University, Seoul 136702, Korea
2    Advanced Forming Research Centre, Inchinnan Drive, Renfrew PA4 9LJ, UK
3    Department of Mechanical Engineering, Indian Institute of Technology Bombay, Mumbai 400076, India
4    Department of Mechanical Engineering, The University of Akron, Akron, OH 44325, USA
*    Correspondence: nayakkanhu83@gmail.com

**Abstract:** Powder metallurgy-based metal matrix composites (MMCs) are widely chosen and used for the development of components in the fields spanning aerospace, automotive and even electronic components. Engineered MMCs are known to offer a high strength-to-weight ($\sigma/\rho$) ratio. In this research study, we synthesized cylindrical sintered samples of a ceramic particle-reinforced aluminum metal matrix using the technique of powder metallurgy. The samples for the purpose of testing, examination and analysis were made by mixing aluminum powder with powders of silicon carbide and aluminum oxide or alumina. Four varieties of aluminum composite were synthesized for a different volume percent of the ceramic particle reinforcement. The hybrid composite contained 2 vol.% and 7 vol.% of silicon carbide and 3 vol.% and 8 vol.% of alumina with aluminum as the chosen metal matrix. Homogeneous mixtures of the chosen powders were prepared using conventional ball milling. The homogeneous powder mixture was then cold compacted and subsequently sintered in a tubular furnace in an atmosphere of argon gas. Five different sintering conditions (combinations of temperature and sintering time) were chosen for the purpose of this study. The density and hardness of each sintered specimen were carefully evaluated. Cold compression tests were carried out for the purpose of determining the compressive strength of the engineered MMC. The sintered density and hardness of the aluminum MMCs varied with the addition of ceramic particle reinforcements. An increase in the volume fraction of the alumina particles to the Al/SiC mixture reduced the density, hardness and compressive strength. The sintering condition was optimized for the aluminum MMCs based on the hardness, densification parameter and cold compressive strength. The proposed powder metallurgy-based route for the fabrication of the aluminum matrix composite revealed a noticeable improvement in the physical and mechanical properties when compared one-on-one with commercially pure aluminum.

**Keywords:** metal matrix composite; powder metallurgy; densification; compaction load; hardness

## 1. Introduction

The powder metallurgy (PM) method is a potentially viable, energetically efficient and economically cost-effective and feasible method for producing both simple and complex parts to the required dimensions. In recent years, the PM technique has been proven to have an edge over conventional casting processes to produce metal matrix composites (MMCs) with the reinforcement being ceramic particles [1–3]. Powder metallurgy-based metal matrix composites (MMCs) are currently being chosen and used for the development of components in the fields spanning aerospace, automotive and even electronics, to name a few [4]. Recently, PM techniques have also been used in additive manufacturing [5]. Powder metallurgy (PM)-based composites have an observably lower density and a higher hardness coupled with a higher porosity when compared with stir casting-based composites. However, the reinforcements are uniformly distributed in the metal matrix using

the powder metallurgy (PM) method when compared with conventional stir casting processes [6]. That is, a near uniform distribution of the reinforcing particles in the metal matrix is not controlled by the conventional casting process, which often results in inhomogeneous physical and mechanical properties of the chosen composite material. In a casting-based Al/SiC composite, an agglomeration of the reinforcing silicon carbide (SiC) particles often results in a weaker bonding with the aluminum matrix [7].

Aluminum and its alloys are usually chosen as the matrix primarily because of its low density, high strength-to-weight ($\sigma/\rho$) ratio, excellent corrosion resistance, ductile matrix and, importantly, a reasonable and economically viable cost. Silicon carbide and alumina ($Al_2O_3$) are the commonly chosen and used reinforcements for aluminum and other metal matrixes due to a combination of favorable properties, including the following: (i) a high service temperature; (ii) a high hot strength; (iii) a high elastic modulus; and (iv) good thermal shock resistance. In addition, they have the tendency to form a strong interfacial bond with the aluminum metal matrix and have a high wear resistance [8–11].

Several studies have been carried out on single reinforcement aluminum/silicon carbide (Al/SiC) composites using the powder metallurgy process for the purpose of evaluating their physical and mechanical properties [12–15]. The porosity and sintered properties of the PM-based composite can be controlled by controlling the sintering parameters, namely: (i) the temperature and (ii) the sintering time [16–18]. The reinforcement particle size and its volume fraction in the Al matrix have a vital role in the density and mechanical properties of Al metal matrix composites [19–22]. The synergetic effect of multiple reinforcements such as $SiC/Al_2O_3$ in an Al matrix alters the porosity that causes the variation in the mechanical properties of Al metal matrix composites [23,24]. Recently, hybrid metal matrix composites have attracted attention for use in different applications and research has been carried out to understand the various properties that these hybrid composites have to offer [25–31].

Zhang et al. (2020) [25] studied the evolution of the microstructure of two varieties of hybrid composites such as Al/SiC/graphite and Al/SiC/graphene by the PM process, where graphene in the Al and nanostructure composite could be formed. In the PM process, the effect of the sintering temperature and compaction pressure on an $Al/Al_2O_3/WS2$ hybrid composite was investigated by Biswal and Sahoo (2020) [27]. A compaction pressure of 560 MPa and a sintering temperature of 600 °C for the $Al/Al_2O_3/WS2$ hybrid composite revealed good densification (porosity ~8%), resulting in a hardness value of 6364 Hv. In a hybrid composite, the volume percentage of the reinforcement also influences its overall properties. Daha et al. (2021) [28] analyzed the contribution of the wt.% of reinforcements on the tribological properties and microhardness of an $Al/Al_2O_3/RGO$ hybrid composite. In their study, a higher wt.% combination of RGO (>0.3 wt.%) with $Al_2O_3$ ($\geq$10 wt.%) decreased the microhardness and tribological effect due to the agglomeration of the reinforcements. Manohar et al. (2022) [29] investigated the microstructure and mechanical properties of Al7075/graphite/SiC hybrid composites; they found that 8 vol.% of graphite and 2 vol.% of SiC maximized the mechanical properties. Proitte et al. (2022) [30] studied the damping behavior of a fiber composite in aeronautical technology. In addition to ceramic reinforcements, iron oxide ($Fe_2O_3$) is also used to make hybrid composites [31]. Moreover, the hybrid composite has a complex constitution with multiple reinforcement particles. The surface area of the specimen, volume fraction, size and sintering conditions greatly influence the overall properties of hybrid composites. Among these, the surface area of the sample is affected by the compaction pressure, resulting in non-homogeneous densification due to die wall–powder interface friction. Therefore, it is necessary to investigate the porosity level, density and hardness as a function of the sintering time and elastic modulus of the part. In this research study, we carefully synthesized cylindrical test specimens with a different part modulus using the conventional powder metallurgy process. The physical and mechanical properties (hardness and compressive strength) of the sintered aluminum composite material were investigated for a different part modulus and sintering time. An exhaustive examination of the microstructure was carried out to

study and concurrently establish the nature of the interface bonding between the hard, elastically deforming ceramic particle reinforcement and the soft and ductile metal matrix. In addition to this, the porosity of the sintered specimens was determined.

## 2. Experimental Approach

Commercially available aluminum powder (Fe: 0.1%; Mn: 0.02%; Ti: 0.03%; Cu: 0.02%; Si: 0.1%; and balance Al) with a mean particle size of 46.16 μm was used for the preparation of the test specimens. In addition, laboratory-based non-metallic ceramic particles such as alumina (aluminum oxide ($Al_2O_3$)) and silicon carbide (SiC) were blended with the aluminum powder to make four varieties of composite materials, each containing a different volume fraction of the ceramic particle reinforcements. The blending process was carried out using a planetary ball mill (FRITSCH, Pulverisette 6) at 100 rpm for 30 min with a zirconium ball (5 mm diameter) to a powder ratio of 10:1. The properties of the chosen metal matrix and the two chosen reinforcements are summarized in Table 1. The # indicates the values that were experimentally determined and * indicates the properties from the supplier (for the Al and $Al_2O_3$ powder (Loba Chemie Pvt. Ltd., Mumbai, India) and for SiC (Alfa Aesar, Mumbai, India)). The chemical composition of the chosen materials is provided in Table 2.

**Table 1.** Properties of the consumables (#: measured at particle level, *: measured at sample level after sintering).

| Material | Mean Particle Size (μm) # | Young's Modulus (GPa) * | Tensile Strength (MPa) * | Coefficient of Thermal Expansion ($10^{-6}$ K$^{-1}$) * | Density (g/cm$^3$) * |
|---|---|---|---|---|---|
| $Al_2O_{3p}$ | 1.27 | 360–400 | 250–300 | 8.5 | 3.95 |
| $SiC_p$ | 8.13 | 400–440 | 310 | 4.8 | 3.2 |
| Al | 46.16 | 70 | 200 | $22.2 \times 10^{-4}$ | 2.7 |

**Table 2.** Composition of the aluminum metal matrix composite.

| Material | Composites | Volume Fraction (%) | Weight for 10 g of Aluminum |
|---|---|---|---|
| Al MMCs (Al/SiC/$Al_2O_3$) | Al/2–SiC/3–$Al_2O_3$ | 2 and 7 vol.% of SiC | 0.375 SiC + 0.692 $Al_2O_3$ |
| | Al/2–SiC/8–$Al_2O_3$ | | 0.396 SiC + 1.950 $Al_2O_3$ |
| | Al/7–SiC/3–$Al_2O_3$ | 3 and 8 vol.% of $Al_3$ | 1.387 SiC + 0.731 $Al_2O_3$ |
| | Al/7–SiC/8–$Al_2O_3$ | | 1.468 SiC + 2.065 $Al_2O_3$ |

The homogeneous powder mixture was cold compacted using a cold uniaxial press with a metallic die with an inner diameter of 10.2 mm. A mixture of 1 g, 2 g and 3 g ($\pm$ 0.080) of the powder mixture was taken for the purpose of compaction to produce a solid with an H/D ratio of 0.5, 1.0 and 1.5, respectively. The homogeneous powder mixture was initially pressed in a cold uniaxial pressing machine (make: SOILLAB; type: hydraulic) to obtain "green" cylindrical test samples that had an outer diameter of 10 mm. The compacted samples were subsequently sintered in a tubular furnace in an atmosphere of argon gas. The sintering cycles are shown in Figure 1; those used for making the hybrid Al MMCs are listed in Table 2. The sintering temperatures were the furnace temperatures where one (700 °C) was near to the melting point of Al. Three different sintering times, 30 min, 60 min and 90 min, at a furnace temperature of 700 °C were selected for the purpose of sintering in order to both study and establish its influence on the density, hardness and microstructure evolution for the Al/7 vol.% SiCp. The sintered Al/7 vol.% SiCp composites that had a different modulus (m = 1.25, 1.67 and 1.87) are shown in Figure 2. The composites (Al/7 vol.% SiC) shown in Figure 2 were used for the purpose of the measurement of the density, hardness and microstructure to examine the modulus effects. The sintered Aluminum MMCs, as shown in Table 2, were further compressed to a 50% reduction in height using a tensile test machine (model: Tinius Olsen, 50 KT). In addition, these composites were also

used for the measurement of hardness and density. The hardness of all the Al MMCs was measured using a Rockwell hardness (HR) tester (RASN-T, 15 T, 1/16 inch ball indenter).

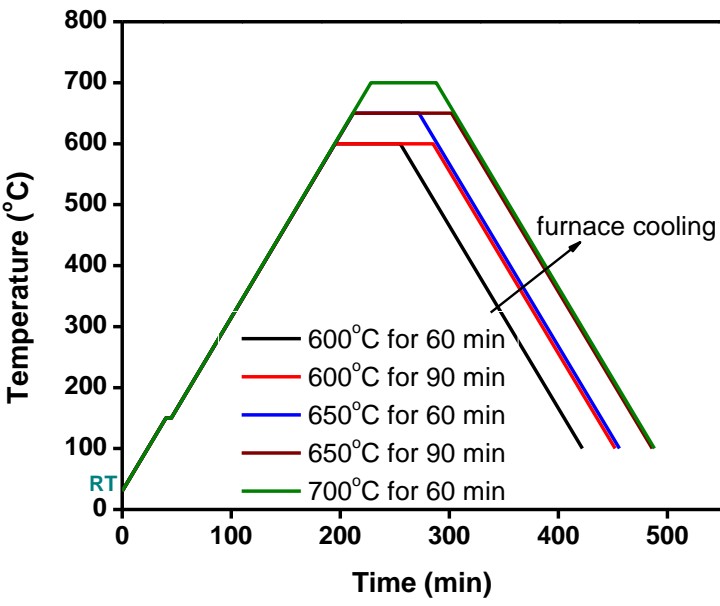

**Figure 1.** Sintering cycles for the aluminum metal matrix composites (Al MMCs).

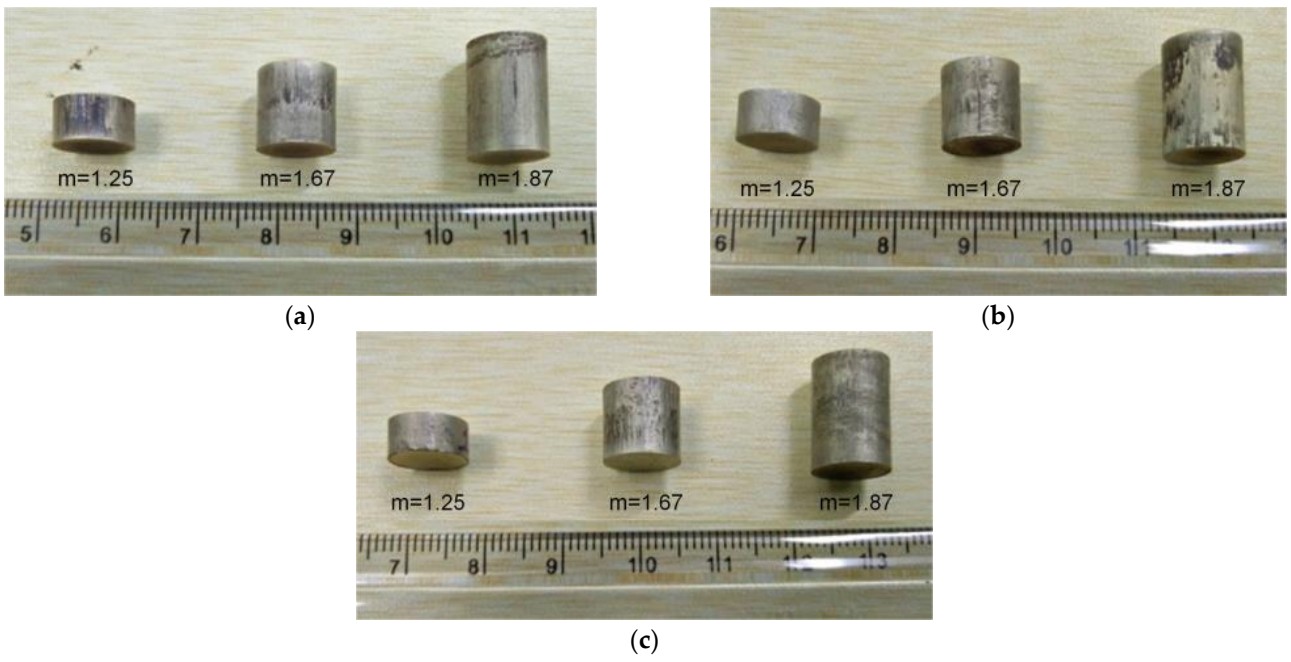

**Figure 2.** Powder metallurgy (PM)-based Al/7 vol.% SiCp composites (m = 1.25, 1.67 and 1.87) sintered at 700 °C for (**a**) 30 min, (**b**) 60 min and (**c**) 90 min.

## 3. Results and Discussion

Subsequent to compaction, the "green" density of the compact aluminum matrix composite (AMC) samples were determined by measuring their mass (to an accuracy of ± 0.01 g) and volume. Both the height (H) and diameter (D) of the samples were measured immediately following compaction (to an accuracy of ± 0.02). The sintered density of the

AMC samples was measured using the principle of Archimedes. The sintered density was calculated in accordance with the relationship:

$$\rho_{AMCs} = \frac{\rho_l \times w_{AMCs}}{\Delta w_{AMCs}}. \tag{1}$$

In this expression, $\rho_l$ is the density of liquid, $w_{AMCs}$ is the weight of the sintered aluminum matrix composite (AMC) in the laboratory air and $\Delta w_{AMCs}$ is the difference in the weight of the sintered AMC between the laboratory air and liquid.

The variations in the sintered density of the Al hybrid composite for the different sintering conditions is shown in Figure 3. As the density of the reinforcing SiCp and $Al_2O_3$p is inherently higher than that of aluminum, its presence in the material led to a higher density of the aluminum matrix composite (AMC), which could be analyzed using the rule of mixtures theory [4]:

$$(\rho_c) = v_r \times \rho_r + (1 - v_r) \times \rho_m. \tag{2}$$

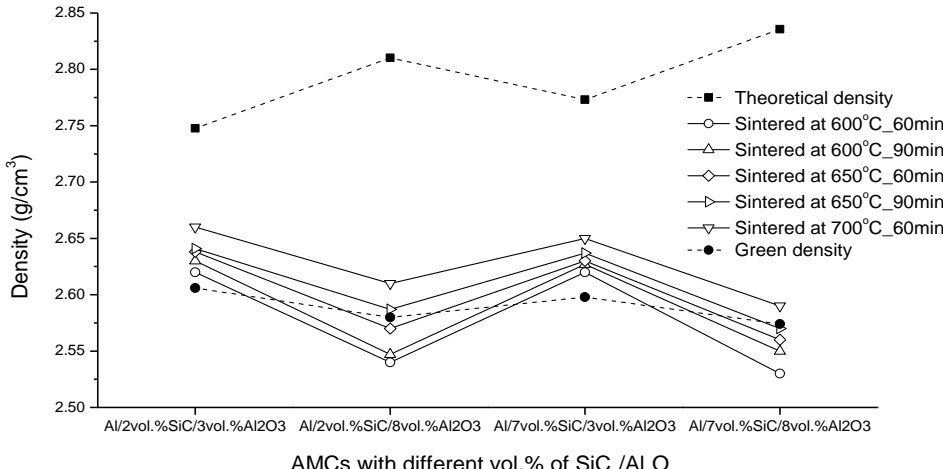

**Figure 3.** Effect of addition of $SiC_p/Al_2O_{3p}$ on density for the different sintering conditions.

In Equation (2), $\rho_c$, $\rho_r$ and $\rho_m$ denote the density of the composite, the density of the particulate reinforcement and the density of the matrix. $v_r$ is the volume percent of the particulate reinforcement. We concluded that the density of the engineered aluminum composite varied between the density of the matrix and the density of the particulate reinforcement. Theoretically, this was true, but experimentally the "green" and sintered density depended upon the conjoint and mutually interactive influences of the following: (i) the compaction pressure; (ii) the sintering condition; and (iii) the compressibility of the reinforcement particles. Due to this, the "green" and sintered density did not follow the trend shown or observed by the theoretical density of the composites. An increase in the addition of the alumina ($Al_2O_3$) particles to the aluminum tended to decrease the density. An excess in the addition of alumina particle reinforcements tended to cause a matrix swelling of the test samples subsequent to sintering.

The "green" density did not follow the theoretical density for the addition of SiCp and $Al_2O_3$p to the aluminum metal matrix. It was evident that the "green" density of the aluminum matrix composite (AMC) compact actually decreased with an increase in the content of both the SiCp (>7 vol.%) and $Al_2O_3$p (>3 vol.%) reinforcing particulates. This is shown in Figure 3. This observation could essentially be attributed to the hard and non-deforming nature of the two chosen particulate reinforcements—i.e., (i) SiCp and (ii) $Al_2O_3$p—which tended to constrict the deformation of the reinforcing particles as well as sliding and possible rearrangement during compaction [5–7]. The sintered density could be improved by increasing both the sintering temperature and time.

The hybrid effect of the reinforcement was studied in the experiments. This effect is shown in Figure 4. The aluminum metal matrix composites (AMCs) containing 2 vol.% $SiC_p$/3 vol.% $Al_2O_3p$ and 7 vol.% $SiC_p$/3 vol.% $Al_2O_3p$ revealed an observable improvement in hardness. However, vice versa was found to be not true. An increase in the volume fraction of the reinforcing $SiC_p$ for a constant volume fraction of $Al_2O_3$ particles contributed to increasing the hardness. An increase in the volume fraction of the alumina ($Al_2O_3$) particles in the composite microstructure reduced the sintered density (Figure 3) with a concomitant decrease in hardness, as shown in Figure 4.

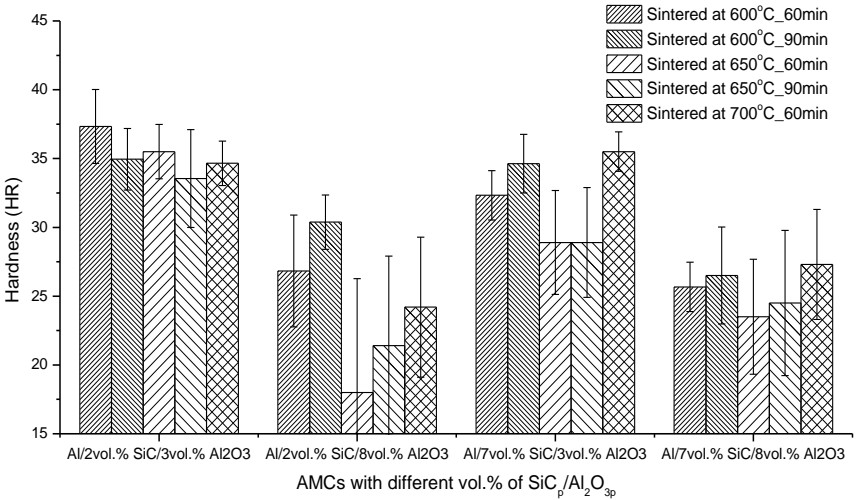

**Figure 4.** Effect of $SiC_p$/$Al_2O_{3p}$ volume fraction on the sintered aluminum matrix composites (AMCs) for the various sintering conditions.

Furthermore, Rockwell hardness tests were carried out on a flat surface of the cylindrical sintered Al/7 vol.% $SiC_p$ specimens with a different modulus (Figure 5a). A similar trend in the variation (as observed for densification) of the hardness is shown in Figure 5b with respect to the modulus of the part. Few samples with a 1.25 modulus and a 1.67 modulus revealed a variation in hardness with an applied load during compaction. This was primarily because of the varying frictional conditions between the reinforcing particles during compaction, which exerted a noticeable level of densification along the length of the chosen part. Uneven interface bonding between the chosen reinforcing $SiC_p$ and Al particles, coupled with the movement of the reinforcement clusters, caused and/or favored an intrinsic variation in the hardness. An improvement in the hardness of the cylindrical samples for a modulus of 1.67 is shown in Figure 5b. This was essentially due to a strong interface bonding between the reinforcing $SiC_p$ particles and the aluminum metal matrix. A decrease in the hardness for a part modulus of 1.87 indicated a lower densification of the aluminum matrix composite. This is also shown in Figure 5b.

The correlation between the hardness and density with respect to the part modulus is shown in Figure 6. The parts sintered for 60 min to 90 min revealed a noticeable improvement in properties (quantified by density and hardness). A high density (2.7 g/cm$^3$) with a low hardness (~34 HR) was observed for a part modulus of 1.87. A higher sintered density of the Al MMCs did not necessarily result in a higher hardness, as in the case of both aluminum and an aluminum alloy. This could be due to the presence of hard reinforcing particles such as $SiC_p$. The correlation was retained for a part modulus of 1.25 and 1.67 and a sintering time of 60 min to 90 min. The sintered density achieved was 2.69 g/cm$^3$ and this was made possible by controlling both the compaction pressure and the sintering parameters; i.e., (i) the sintering temperature and (ii) the time. Moreover, increasing the sintering time and temperature could increase the density and hardness [2].

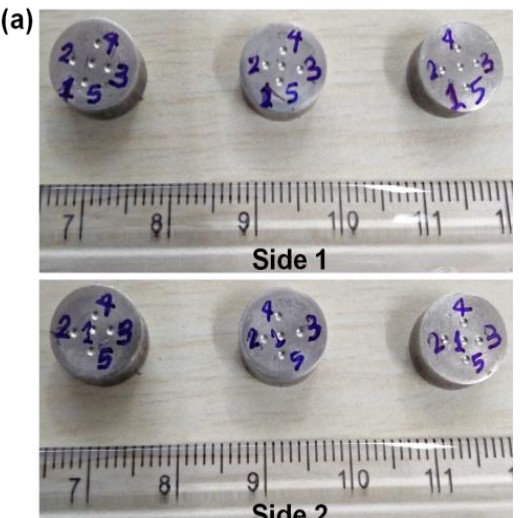

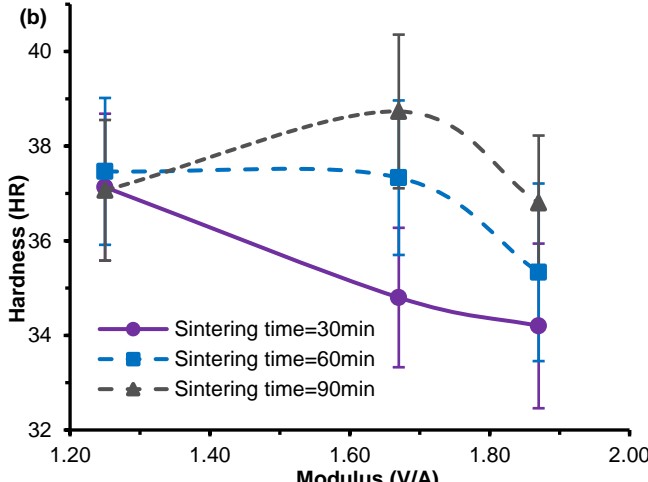

**Figure 5.** Hardness of the sintered aluminum MMCs (sintering temperature: 700 °C): (**a**) location of indentation on the test specimen and (**b**) variation in hardness with modulus.

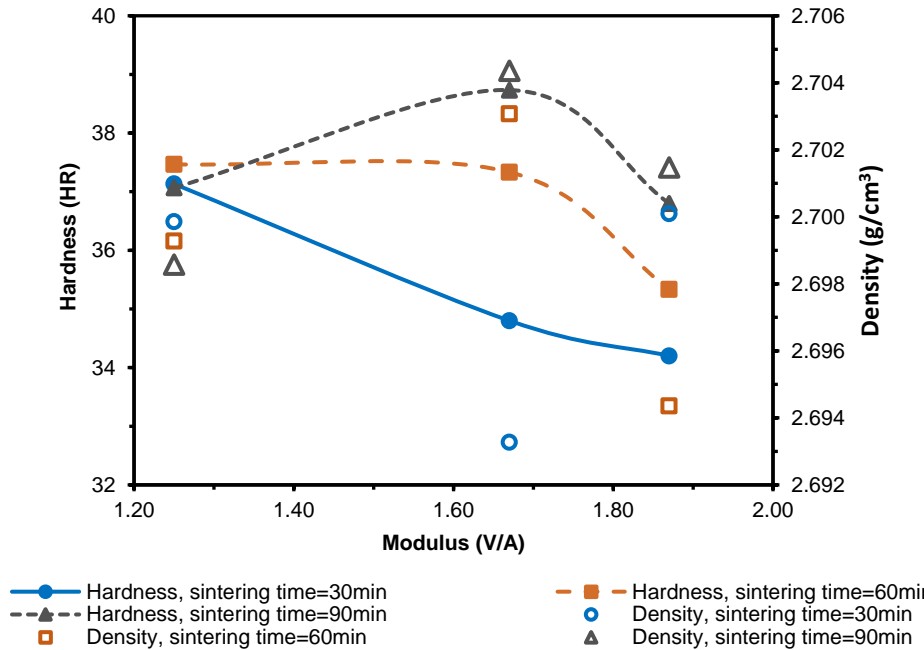

**Figure 6.** Correlation between hardness and density for Al/7 vol.% SiC$_P$ for the different part moduli (sintering temperature: 700 °C).

A metallographic examination and analysis of the Al MMCs was carried out to investigate the effect of changing the sintering temperature and sintering time on the distribution of the reinforcements; i.e., SiC particles in the aluminum metal matrix. Optical micrographs were taken using a LEICA DM2500M microscope. The distribution of the reinforcing SiC particles in both the aluminum matrix and the aluminum grain boundaries is shown in Figure 7 for a part modulus of 1.87.

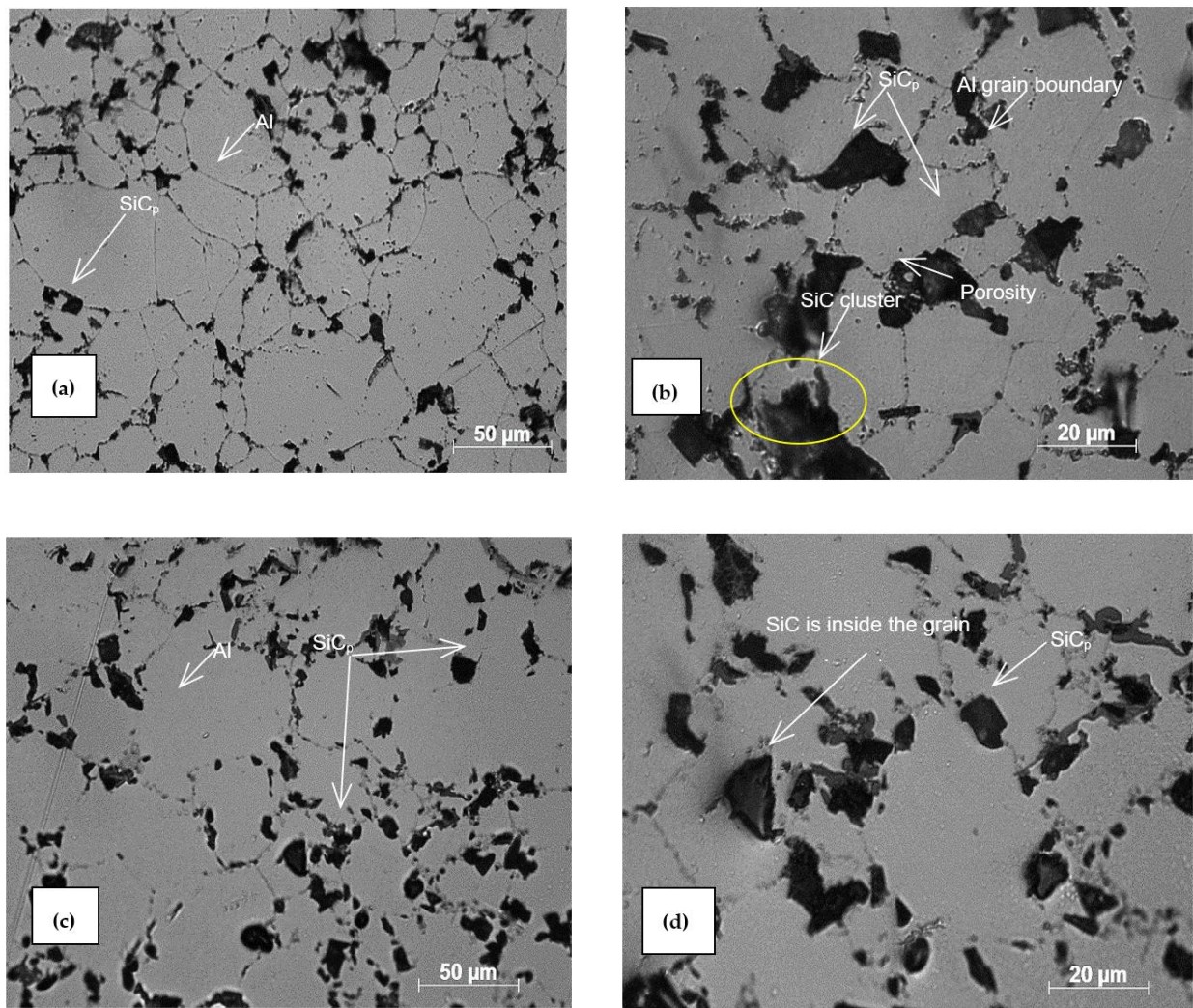

**Figure 7.** Micrographs of the Al/7 vol.% SiC$_p$ MMCs sintered at 700 °C: (**a**) sintering time 30 min, 20× magnification; (**b**) sintering time 30 min, 50× magnification; (**c**) sintering time 90 min, 20× magnification; (**d**) sintering time 90 min, 50× magnification.

The distribution of the reinforcing particulates was found to be changed upon increasing the sintering time. Most of the reinforcing SiC particles were located both at and along the grain boundaries. The reinforcing particles tended to gradually move along the grain boundary of the aluminum due to an increase in grain size following 90 min of sintering. Moreover, an improved distribution of the reinforcing SiC particles (SiCp) resulted in improved density and hardness following sintering for 90 min. The porosity level reduced for the sintered Al/7 vol.% SiC$_p$ with an increase in the sintering time. However, the presence of the clustering of SiC could depend on the degree of homogeneity of the particle dispersion [32].

For the hybrid composite (Al/SiC$_p$/Al$_2$O$_{3p}$), the two chosen reinforcing particles (SiC$_p$ and Al$_2$O$_{3p}$) revealed a combined effect on the compressive strength or stress, as shown in Figure 8. Alumina of 3 vol.% in an aluminum matrix and 3–7 vol.% of SiCp (Al/2 vol.% SiCp/3 vol.% Al$_2$O$_3$p and Al/7 vol.% SiCp/3 vol.% Al$_2$O$_3$p) revealed an improvement in both the compressive strength and ductility prior to a failure by fracture, as shown in Figure 8a,c. The Al/2 vol.% SiCp/8 vol.% Al$_2$O$_3$p and Al/7 vol.% SiCp/8 vol.% Al$_2$O$_3$p MMCs in Figure 8b,d, respectively, show a lower compressive strength. These composites were poorly densified at sintering temperatures of ≤650 °C, indicating an earlier fracture than others. Moreover, a higher volume percentage of reinforcements in the Al matrix

at a lower sintering temperature were not densified, as revealed from their low density (Figure 3) and hardness values (Figure 4).

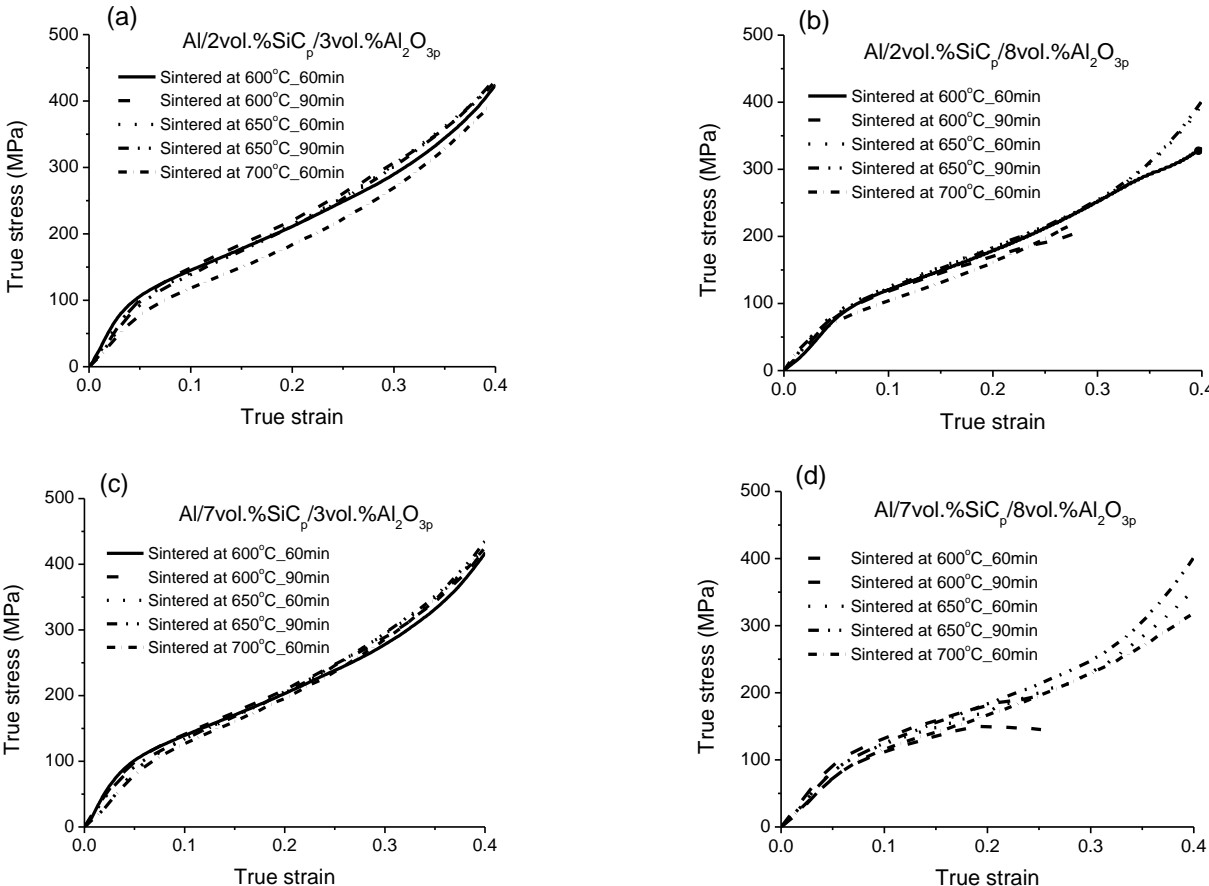

**Figure 8.** True stress versus true strain curves for the $Al/SiC_p/Al_2O_{3p}$ composite for the different sintering conditions.

In the powder metallurgy process, the aluminum particles were surrounded by the reinforcing SiCp and $Al_2O_3$ particles. An excess in the addition of these hard, elastically deforming and brittle ceramic particles restricted their movement during compaction. The diffusion of the aluminum particles and their growth at the region of the neck was restricted by the reinforcing ceramic particles, which tended to lower the sinterability. This could be the reason for the lowering of the compression strength of the Al MMCs with a higher vol.% of reinforcements (>7 vol.% of SiC and >3 vol.% of $Al_2O_3$), as is shown in Figure 8.

The deformed and undeformed samples of the engineered composite material are shown in Table 3. Barreling of the test samples was observed for both the Al/2 vol.% SiC/8 vol.% $Al_2O_3$ and Al/7 vol.% SiC/8 vol.% $Al_2O_3$ MMCs for all the sintering conditions. An increase in the sintering temperature did not influence the true stress due to the low sinterability of the engineered aluminum composites. Fractures on the circumference of the specimens were observed for all sintering conditions, except at a condition of 700 °C and 60 min for Al/2 vol.% SiC/3 vol.% $Al_2O_3$ and Al/7 vol.% SiC/3 vol.% $Al_2O_3$. It was seen that the size of the crack increased with an increasing vol.% of reinforcements. The crack on the circumference was perpendicular to the compression loading, indicating a brittle fracture.

**Table 3.** Deformation of the aluminum matrix composites (AMCs) following various sintering conditions.

| Al MMCs | Sintering Condition | | | | |
|---|---|---|---|---|---|
| | 600 °C/60 min | 600 °C/90 min | 650 °C/60 min | 650 °C/90 min | 700 °C/60 min |
| Al/2 vol.% SiC/3 vol.% Al$_2$O$_3$ | | | | | |
| Al/2 vol.% SiC/8 vol.% Al$_2$O$_3$ | | | | | |
| Al/7 vol.% SiC/3 vol.% Al$_2$O$_3$ | | | | | |
| Al/7 vol.% SiC/8 vol.% Al$_2$O$_3$ | | | | | |

## 4. Conclusions

Two categories of aluminum-based metal matrix composites (Al MMCs) were synthesized by the PM process: a hybrid composite Al/SiC/Al$_2$O$_3$ and an Al/7 vol.% SiC composite. The influence of the part modulus and sintering time on the density and hardness of the Al/7 vol.% SiC composite was investigated at a sintering temperature of 700 °C. In addition, the density, compressive strength and hardness of different Al/SiC/Al$_2$O$_3$ MMCs were examined for various combinations of sintering conditions. The following conclusions were drawn from the current experimental study and analysis.

1. The addition of particulate reinforcements such as SiC and alumina to an aluminum matrix improved the compressive true strength. An excess in the addition of reinforcements (beyond 7 vol.% of SiC and 3 vol.% of Al$_2$O$_3$) caused poor sinterability for the chosen sintering condition.
2. An increase in the sintering time contributed less to the compressive strength of the composite material (Al/SiC/Al$_2$O$_3$ MMCs). However, a higher densification was evident for the sintered density, hardness and metallographic analysis when the sintering time was increased from 30 min to 90 min for the Al/7 vol.% SiC composite.
3. The sintering condition of 700 °C for 60 min was suitable for the aluminum metal matrix composites (Al MMCs) to achieve a higher density and improved mechanical properties for the hybrid composites: (a) Al/2 vol.% SiC/3 vol.% Al$_2$O$_3$ and (b) Al/7 vol.% SiC/3 vol.% Al$_2$O$_3$.
4. A marginal variation in the sintered properties was seen when varying the part modulus. Further, the parts with the 1.67 modulus produced the best combination of sintered properties. Interfacial bonding between the reinforcing SiC particles and the aluminum metal matrix was not observably influenced by changing the part modulus, but was significantly altered by the sintering parameters (sintering temperature and sintering time).
5. At 650 °C and below, the matrix phase and the reinforcement phase did not uniformly transfer the compressive load.

Further research is required for an in-depth microstructural analysis to examine the additional compounds formed in Al MMCs due to a higher sintering temperature and their contribution to the overall properties of Al/SiC/Al$_2$O$_3$ MMCs.

**Author Contributions:** Conceptualization, K.C.N. and K.K.R.; methodology, K.C.N.; software, K.C.N.; validation, K.C.N., K.K.R. and P.P.D.; formal analysis, K.C.N.; investigation, K.K.R.; resources, P.P.D.; data curation, T.S.S.; writing—original draft preparation, K.C.N.; writing—review and editing, K.K.R.; visualization, K.K.R.; supervision, P.P.D.; project administration, T.S.S.; funding acquisition, P.P.D. All authors have read and agreed to the published version of the manuscript.

**Funding:** This research received no external funding.

**Institutional Review Board Statement:** Not applicable.

**Informed Consent Statement:** Not applicable.

**Data Availability Statement:** Not applicable.

**Conflicts of Interest:** The authors declare no conflict of interest.

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
