# Peer review of "Synthesis of an Aluminum Alloy Metal Matrix Composite Using Powder Metallurgy: Role of Sintering Parameters"

_applsci, doi:10.3390/app12178843_

Round 1

Reviewer 1 Report

The Communication deals with Synthesis of an Aluminum Alloy Metal Matrix Compostite using Powder Metallurgy: Role of Sintering Parameters. 

According to the reviewer, the Communication is worth publishing at Applied Sciences Journal, 

but some corrections are needed and then the Communication can be accepted for publication in the journal.

While the authors have made considerable research effort, 

the presentation of the Communication and the results must be proved. 

Additionally make the following corrections to the manuscript:

Comment 1

Lines 2 - 4

Why capitals?

Comment 2

Line 5

Kedarnath K. Rane 2 ,Prashant

The authors should replace

Kedarnath K. Rane 2, Prashant

Line 6

SOUTH KOREA

The authors should replace

South Korea

Line 10

INDIA

The authors should replace

India

Line 15

metal matrix composites [MMCs]

The authors should replace

Metal Matrix Composites [MMCs]

Line 28

study.   The density and hardness of each

The authors should replace (delete the extra spaces)

study. The density and hardness of each

Line 30

MMC.   The sintered

The authors should replace (delete the extra spaces)

MMC. The sintered

Line 31

reinforcements.   An increase 

The authors should replace (delete the extra spaces)

reinforcements. An increase 

Delete the extra spaces in the sentence beginning.

Line 118

for.

The authors should replace

for:

Comment 3

Exteded text editing:

Line 56

the aluminum matrix [4] 

The authors must insert a "."

the aluminum matrix [4]. 

Line 57

.Aluminium

The authors must delete the "."

Aluminium

Line 254

following.various sintering

The authors must delete the "."

following various sintering

Comment 4

Line 86

The authors must give more details for the preperation of the test specimens.

For examble, the authors mension in abstract "using conventional ball milling.", in the mean text there is not an analytical description.

Comment 5

Table 1

The authors must give more details for the values (data from supplier or author's experiments?)

Comment 6

The authors must format the Table 2 according to the journal's instructions (alignment for the rows "Composites" and "Weight for 10 gram of Aluminum").

Comment 7

Figure 1

The authors must insert values in axis x (Time (min)) and y (Temperature (oC).

Comment 8

Lines 123 - 128

Line 168

Figure 7

The authors must give more details for the experimental devices (type, model,...)

Comment 9

While the authors give results, there is no results for 30 minutes.

Comment 10

Figures 5 and 6

The authors must give more details for the results: 

Sintering time=60min Temperature ?

Sintering time=90min Temperature ?

Comment 11

The Figure 7 must be accompanied on the same page as the Figure's title.

Comment 12

Figure 8

The authors must increase the visibility (each case to be clear which line it is) of the Figure 8.

Comment 13

Lines 266 - 283

The authors must format the text (full alignment).

Comment 14

Increase the number of the reference papers including (primarily) from Applied Sciences.

The authors use 0 paper from Appied Sciences journal / 1 papers from MDPI Journals / 19 papers from journals (References)

Τhe number for papers from MDPI journals is considered insufficient (in reviewer's opinion).

Comment 15

The authors must format the References according the journal's instructions (Vol. Italics).

References should be described as follows, depending on the type of work:

Journal Articles:

1. Author 1, A.B.; Author 2, C.D. Title of the article. Abbreviated Journal Name Year, Volume, page range.

Comment 16

Ref. [1] and [2]:

The title of the papers is missing.

Reviewer 2 Report

Journal: Applied Sciences (ISSN 2076-3417)

Manuscript ID: Applied Sciences-1867042

The authors presented a communication about “Synthesis of an Aluminum Alloy Metal Matrix Composite using Powder Metallurgy: Role of Sintering Parameters”. I think the communication is well organized and appropriate for the “Applied Sciences” journal, but the communication will be ready for publication after major revision.

·         A sentence about numerical results should be added to the abstract.

·         For the introduction, please add more current references and briefly explain them. The following references can be added.

https://doi.org/10.3390/ma14154217

https://doi.org/10.3390/nano12132154

https://doi.org/10.1016/j.jmrt.2021.11.114

https://doi.org/10.1016/j.ceramint.2021.07.165

·         The melting temperature of aluminum is 660 °C. In your experiments, you went up to the sintering temperature of 700 °C. Liquid phase sintering did not occur?

·         In Figure 7, item no (a, b) is missing. (Line 224-225)

·         Results and discussion and conclusion parts are inadequate according to citation and analyze in detail. There should be the importance of the study in detail, comparison results with other approaches in literature, the success of the experimental results.

·         Improve the results and discussion and conclusion parts.

·         Please fix the typographical and eventual language problems in paper.

·         The paper is well-organized yet there is a reference problem. First, your reference list contains no paper from “Applied Sciences” journal. If your work is convenient for this journal’s context then there are many references from this journal. Secondly, cited sources should be primary ones. Namely, indexed area shows the power of a paper and directly your paper’s reliability. Please make regulations in this direction.

·         The article should be rearranged by taking into account the journal writing rules and citation rules.

*** Authors must consider them properly before submitting the revised manuscript. A point-by-point reply is required when the revised files are submitted.

Round 2

Reviewer 1 Report

Lines 117 - 120

The symbols (# and *) are missing from the Table 1.

Lines 79 - 83

In the Pm process, replace In the PM process,

temperature of 600oC for (o format)

References

Format - same text size

Full alignment

Reviewer 2 Report

The authors made all requested corrections. I believe this article may be published in its final form in "Applied Sciences."